# Learning Long-Horizon Action Dependencies in Sampling-Based Bilevel Planning

**Bartłomiej Cieślar**[1], **Leslie Kaelbling**[1], **Tomás Lozano-Pérez**[1], and **Jorge Mendez-Mendez**[2]

[1]CSAIL MIT      [2]Stony Brook University

{bcieslar,lpk,tlp}@csail.mit.edu    jorge.mendezmendez@stonybrook.edu

**Abstract:** Autonomous robots will need the ability to make task and motion plans that involve long sequences of actions, e.g. to prepare a meal. One challenge is that the feasibility of actions late in the plan may depend on much earlier actions. This issue is exacerbated if these dependencies exist at a purely geometric level, making them difficult to express for a task planner. Backtracking is a common technique to resolve such geometric dependencies, but its time complexity limits its applicability to short-horizon dependencies. We propose an approach to account for these dependencies by learning a search heuristic for task and motion planning. We evaluate our approach on five quasi-static simulated domains and show a substantial improvement in success rate over the baselines.

**Keywords:** task and motion planning, long-horizon, learning for planning

## 1 Introduction

Solving long-horizon problems, (e.g., preparing a meal), is key toward deploying autonomous robots. Model-based approaches, and in particular task and motion planning (TAMP) [1], offer strong generalization to handle such tasks. A major challenge is that the decisions made early on may be incorrect, but this may only become apparent multiple steps later, such as picking a cucumber in the middle and later needing to slice it down the middle. We propose an approach that learns how to use an existing world model to plan *efficiently*, by predicting the long-horizon consequences of the continuous choices made early during planning.

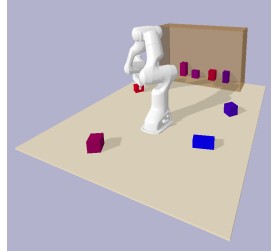

Figure 1: Packing domain from Section 5. The goal is to place blocks in the box, but space constraints require early placements to leave room for later ones.

Bilevel TAMP approaches [2, 3, 4, 5, 6] split robot decision-making into a task planner, which produces a sequence of abstract actions (e.g., pick→ ··· →slice), and a motion planner, which computes motions to enact the abstract actions. If all of the long-horizon dependencies are encoded at the task-planner level (e.g., picking a cucumber requires the gripper to be free, and picking a knife causes the gripper to *not* be free), then TAMP can find a valid sequence of actions with classical planning systems such as Fast Downward [7]. However, long-horizon dependencies that arise from purely geometric constraints are difficult to encode at the task-planner level. One solution for such cases is to find a sequence of abstract actions, and then use a combination of sampling (to determine continuous parameters for executing each abstract action) and backtracking (to recursively resample parameters to resolve all dependencies). While backtracking has succeeded at solving a variety of robot problems [6, 5, 8], its runtime is exponential in the range of the dependencies between actions and often relies on hand-crafted sampling distributions to handle situations where the set of valid samples has small measure relative to the unconstrained parameter space. As our experiments show, this makes seemingly simple problems intractable for bilevel TAMP approaches.

We propose a mechanism that learns to reject unpromising samples (e.g., an incorrect cucumber grasp) by foreseeing the feasibility of further search. Our approach efficiently collects data of successful and failed plans, and trains a transformer to predict whether a state is conducive to a solution, given a chosen sequence of abstract actions. We show empirically that our method substantially im-

8th Conference on Robot Learning (CoRL 2024), Munich, Germany.

proves the efficiency of backtracking, resulting in a much higher success rate given a time limit. Our contributions include: 1) a formalization of long-horizon action dependencies, 2) a transformer architecture for estimating the likelihood of finding parameters for a task plan, 3) an efficient data collection scheme for the classifier, and 4) a modified backtracking method guided by the classifier.

## 2  Related Work

Many recent works have developed learning mechanisms to improve the capabilities of model-based planning. Some methods learn planning *models*, in the form of state abstractions, abstract actions, controllers, or low-level transition functions [6, 8, 4, 9, 10, 11, 12]. The learned models are then used as part of the planning domain itself, to perform planning. Our approach instead learns mechanisms to improve the efficiency of planning given a known model. Within this space, existing approaches have learned parameter samplers for backtracking within bilevel TAMP, using either Gaussian [8, 6] or diffusion-based samplers [5]. Prior work has shown how to combine model learning and sampler learning into an integrated learning approach [8, 6]. Other works focus on utilizing existing imperfect domain models for planning by avoiding discrepancies with the real world [13].

Related to our work, which aims to predict motion planning failures before they occur, are the works on culprit detection problem, focusing on explaining failures of motion planning [14, 15, 16, 17, 18]. Within those, the work most closely related to ours uses a learned culprit detector to backtrack the motion planning multiple steps at a time (backjump) [18]. Our approach uses this backjumping procedure as a fallback for cases when the feasibility classifier produces false positives, using the confidence from our model to determine the most promising number of backtracking steps. Compared to [18], our approach addresses the challenge of efficiently gathering data by jointly learning the backtracking optimization heuristic and generating the data.

Non-model-based approaches to construct long-horizon plans have included sampling trajectories from diffusion models [19] or performing gradient descent [20]. Because these mechanisms lack a model to predict state changes resulting from the robot's actions, they struggle with generalization. Other works explore long-horizon feasibility analysis for task plans, by learning a heuristic to aid task planners [21, 22] or using a learned model directly as a planner [23]. These methods are complementary to our work and could be used in combination with it.

## 3  Problem Formulation

### 3.1  TAMP Formulation

A deterministic TAMP domain $\mathcal{Z} = \langle O, T, \Lambda, \bar{\Omega}, \phi, S, A, f, \alpha \rangle$ consists of a set of objects $o \in O$ (e.g., the red mug) of types $t \in T$ (e.g., mugs). The low-level elements are a set of object-oriented *continuous states* $s \in S = \prod_{o \in O} S_o$, a set of *continuous actions* $a \in A$, and a partial *low-level transition function* $f : S \times A \to S \cup \{\bot\}$ ($\bot$ denotes execution failure). Objects of a type share a common state set: $S_t \supset S_o$ (e.g., pose and size of the red mug). The abstract elements are the set of *abstract states* $\lambda \in \Lambda$ (e.g., the mug is **red** and **on the table**, the table is **on the floor**) and the set of *lifted abstract actions* $\bar{\omega} \in \bar{\Omega} : t_1 \times \ldots \times t_{m_{\bar{\omega}}} \to \Omega$ (e.g., pick$\langle$object$\rangle$), which can be grounded into *ground abstract actions* $\omega = \bar{\omega}(o_1 \in t_1, \ldots, o_{m_{\bar{\omega}}} \in t_{m_{\bar{\omega}}}) \in \Omega$ (e.g., pick$\langle$red mug$\rangle$) with specific typed objects. The partial *abstract transition function* $\phi : \Lambda \times \Omega \to \Lambda \cup \{\bot\}$ specifies how ground abstract actions change the abstract state (e.g., picking the red mug removes it from the table).

The state *abstraction function* $\alpha : S \to \Lambda$ relates the continuous and abstract states (e.g., given the red mug's pose, it is on the table), while abstract and continuous actions are related by *controllers* $\mathfrak{c}_\omega : \Theta_{\mathfrak{c}_\omega} \times S \to S \cup \{\bot\}$ for each ground abstract action $\omega$ (e.g., a **pick object** controller). Each controller is parameterized by $\theta \in \Theta_{\mathfrak{c}_\omega}$ (e.g., a grasp for the object) and follows the constraints of the abstract transition function, so $\forall_{s \in S; \theta \in \Theta_{\mathfrak{c}_\omega}} \mathfrak{c}_\omega(\theta, s) = s' \implies \phi(\alpha(s), \omega) = \alpha(s')$. The controller executes a sequence of actions whose resulting states are governed by the transition function $f$. If successful, the controller results in a state $s'$ that matches the expected abstract state $\alpha(s')$.

A TAMP problem $\langle s_0, G \rangle$ consists of an initial low-level state $s_0$ and a set of abstract goal states $G$ (e.g., the red mug is on the third shelf). One way to solve a TAMP problem is to first find a

task plan $\pi = \langle \omega_1, \ldots, \omega_n \rangle$ of ground abstract actions that induces a sequence of abstract states $\lambda_0, \ldots, \lambda_n$ such that $\alpha(s_0) = \lambda_0$, $\forall_{1 \leq i \leq n} \phi(\lambda_{i-1}, \omega_i) = \lambda_i$, and $\lambda_n \in G$, using search algorithms such as Fast Downward [7] with A* [24]. TAMP solvers then *refine* the task plan into a *controller plan* $p = \langle \theta_1, \ldots, \theta_n \rangle$, which induces states $\sigma = \langle s_0, \ldots, s_n \rangle$ such that $\forall_{1 \leq i \leq n} \mathfrak{c}_{\omega_i}(\theta_i, s_{i-1}) = s_i$.

## 3.2 SeSamE Formulation

SeSamE approaches [5, 6] set the controller parameters using *sampling distributions* $\psi_\omega$ associated with each ground abstract action $\omega$. Samplers then generate controller parameters $\theta \sim \psi_\omega(s)$ using those distributions, conditioned on the continuous state. Parameters are found via backtracking search on an internal model of the domain (Algorithm 1 [2]), which recursively extends the current refinement $\langle \theta_1, \ldots, \theta_i \rangle$. The main loop (Lines 4-11) repeatedly samples new controller parameters for the current ground abstract action $\omega_{i+1}$ (Line 5), checks if the sampled parameters enable computing the next state that matches the expected abstract state (Lines 6-8), and continues the search recursively (Lines 3, 9-11). If there exists a dependency between ground abstract actions $\omega_i$ and $\omega_{i+d}$, the algorithm will take $\exp(O(d))$ steps to detect if $\omega_i$ is sampled incorrectly. Note that Algorithm 1 is not complete (it can become probabilistically complete if we allow the first action's parameters $\theta_1$ to be resampled *ad infinitum*), and the base of the exponent depends on $r_{\text{iter}}$.

```
1: function BT(s_i, i, π)
2:    π : ⟨ω_1, ..., ω_n⟩
3:    if i = n then return ⟨⟩
4:    for 1 ... r_iter do
5:        θ_{i+1} ∼ ψ_{ω_{i+1}}(s_i)
6:        s_{i+1} := c_{ω_{i+1}}(θ_{i+1}, s_i)
7:        if s_{i+1} = ⊥ then
8:            continue
9:        p := BT(s_{i+1}, i+1, π)
10:       if p ≠ ⊥ then
11:           return p ⊕ ⟨θ_{i+1}⟩
12:   return ⊥
```
Algorithm 1: Classic Backtracking.

## 3.3 Graphical View of Long-Range Dependencies

To see why Algorithm 1 struggles with long-range dependencies, consider the (infinite, layered, and directed) graph of continuous states reachable from the initial state $s_0$ after any number $i$ of steps, following a task plan $\pi = \langle \omega_1, \ldots, \omega_n \rangle$. We will characterize the states $s_i$ that enable reaching states $\langle s_{i+1}, \ldots, s_n \rangle$ that match the abstract states $\langle \lambda_{i+1}, \ldots \lambda_n \rangle$ associated with $\pi$. Our approach in Section 4 learns to detect states that don't satisfy this condition and skips them during search.

At a high level, we want to determine whether there exists a further task plan refinement from each continuous state $s_i$ and task plan prefix length $i$. This condition can be defined inductively as: the state $s_i$ matches the expected abstract state $\lambda_i$, and some continuous state $s_{i+1}$ reachable from $s_i$ by following the next action in the plan $\omega_{i+1}$, permits refining the task plan. Formally, we consider the graph $\langle V, \hookrightarrow \rangle$. $V_i$ at each layer $i$ is defined inductively. The base case contains only the initial state: $V_0 \triangleq \{\langle s_0, 0 \rangle\}$. The $(i+1)$-th layer contains all the states that can be reached by following the $(i+1)$-th action, $\omega_{i+1}$, from any state in the previous layer $V_i$: $V_{i+1} \triangleq \{\langle s', i+1 \rangle : \exists_{\langle s, i \rangle \in V_i; \theta \in \Theta_{\mathfrak{c}_{\omega_{i+1}}}} \mathfrak{c}_{\omega_{i+1}}(\theta, s) = s'\}$. The nodes in the graph are $V \triangleq \bigcup_{0 \leq i \leq n} V_i$. The edge relation $\hookrightarrow$ denotes that there is a parameterization of the $(i+1)$-th controller that brings a state $s$ to a next state $s'$: $\langle s, i \rangle \hookrightarrow \langle s', i+1 \rangle \triangleq \exists_{\theta \in \Theta_{\mathfrak{c}_{\omega_{i+1}}}} \mathfrak{c}_{\omega_{i+1}}(\theta, s) = s'$. The condition for refinability of the task plan from a given state is defined inductively. In the base case, $\gamma(\langle s, n \rangle) \triangleq \alpha(s) = \lambda_n$. The inductive step is the cause for the computational difficulty of backtracking in SeSamE:

$$\gamma(\langle s, i \rangle) \triangleq \underbrace{\alpha(s) = \lambda_i}_{\beta(\langle s, i \rangle)} \wedge \underbrace{\exists_{\langle s', i+1 \rangle \in V} \gamma(\langle s', i+1 \rangle) \wedge \langle s, i \rangle \hookrightarrow \langle s', i+1 \rangle}_{\eta(\langle s, i \rangle)} . \tag{1}$$

To illustrate, in Figure 2 we show the structure of the Donut domain from Section 5, where a robot picks a donut, adds toppings using topping machines, and places the donut in a box or on a shelf (depending on the goal, see Figure 4 for visualization). The donut must be grasped from the top to place it in the box, and from the side for the shelf; this is the main point of failure. For now, assume that the goal is to place the donut in the box. For this case, $\gamma$ holds for all states in the top row ($1A$–$6A$) corresponding to top grasps, and does not hold for states in the bottom row ($1B$–$6B$). However, Algorithm 1 uses only $\beta$ to decide whether to continue expanding the current refinement. In our example, $\beta$ holds for states in the bottom row ($1B$–$6B$), which are not conducive to a successful box placement. However, using backtracking to approximate $\eta$ takes exponentially long in $n - i$. If the donut is grasped from the side, this causes the search to focus on retrying topping actions for a

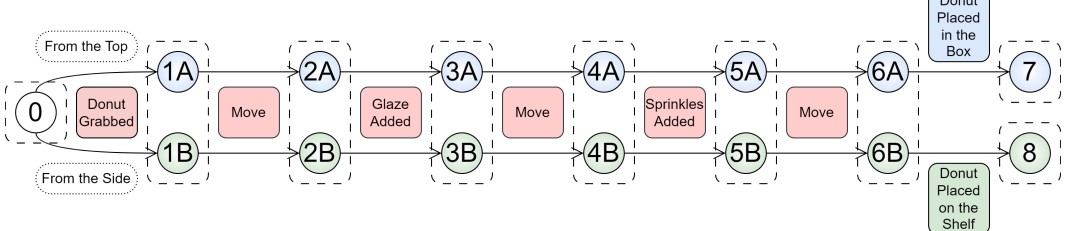

Figure 2: The representation of the graph of continuous states of the Donut domain from Section 5. The robot picks up a donut, either from the side or from the top (which is not encoded in the abstract state), moves to and uses different topping machines, and then places the donut either in the box (state 7) or on the shelf (state 8). The donut cannot be placed in the box if it's grasped from the side and vice versa. The colored boxes describe the ground abstract actions $\omega$ available to the robot, with the arrows signifying corresponding continuous actions (achievable by the controller for a given ground abstract action). The circles mark the groups of similar continuous states, while the dashed rectangles group them into abstract states.

very long time (reaching Line 8 in Algorithm 1 each time the robot fails to execute a box placement and reach state 7), ignoring that the culprit of failure was the initial grasp.

One way to make this process more efficient would be to evaluate $\eta(\langle s_{i+1}, i+1 \rangle)$ before Line 9. In other words, we want to assess for each state $s_{i+1}$ and plan suffix $\omega_{i+1}, \ldots \omega_n$, whether there exists a valid refinement of the suffix. We could then evaluate this condition before Line 9 and, if it doesn't hold, return to Line 5 to resample parameters $\theta_{i+1}$. This would avoid the expensive backtracking necessary for Algorithm 1 to discover that the choice of $\theta_{i+1}$ was incorrect. In practice, a method that leverages this should only consider sets of states with non-zero measure of being reached via refinement under the sampling distribution $\psi_{\omega_{i+1}}$, since otherwise the planner would not find them.

## 4 Learning to Optimize the Backtracking Search

To reduce the backtracking, we propose to learn $\eta$ to determine whether it is possible to further refine the controller plan from the current state $s_i$ and use it as a heuristic for search. We do so using a *feasibility classifier* $h$ that, based on a sequence of prior states $\sigma = \langle s_0, \ldots, s_i \rangle$ and the task plan $\pi = \langle \omega_1, \ldots, \omega_n \rangle$, classifies whether a given task plan is likely to be refinable from the current state. See Appendix E for a simplified explanation of this section.

### 4.1 Backtracking with Feasibility Classifier

Algorithm 2 uses the classifier to prune controller parameter samples that would lead to unrefinable states. With each of $r_{\text{iter}}$ search attempts (Lines 4-14), the algorithm first samples the next controller parameters $\theta_{i+1}$ (Line 5) and tries to run the controller to produce the next state $s_{i+1}$ (Lines 6-7). Then, it uses the feasibility classifier to predict whether further search is likely to succeed from that state (Lines 8-9). The search then continues recursively (Lines 3, 10-12). Inspired by [18], if the current search step fails, the algorithm backtracks to the previous step with the lowest feasibility classifier confidence $\text{conf}_{\text{min}}$ (Lines 13-14). This *backjumping* procedure further reduces backtracking by resampling parameters that are least likely to succeed, under the learned classifier.

```
1: function BT-h(σ, π, conf_min, h)
2:     σ : ⟨s_0, ..., s_i⟩, π : ⟨ω_1, ..., ω_n⟩
3:     if i = n then return ⟨⟩
4:     for 1 ... r_iter do
5:         θ_{i+1} ~ ψ_{ω_{i+1}}(s_i)
6:         s_{i+1} := c_{ω_{i+1}}(θ_{i+1}, s_i)
7:         if s_{i+1} = ⊥ then continue
8:         η, conf := h(π, σ ⊕ ⟨s_{i+1}⟩)
9:         if ¬η then continue
10:        p := BT-h(σ ⊕ ⟨s_{i+1}⟩, π,
                     min(conf_min, conf), h)
11:        if p ≠ ⊥ then
12:            return p ⊕ ⟨θ_{i+1}⟩
13:        if conf > conf_min then
14:            return ⊥
15:    return ⊥
```

Algorithm 2: Backtracking with Feasibility Classifier. The blue parts highlight changes from Figure 1.

### 4.2 Feasibility Classifier Architecture

Per the definition in Equation 1, our classifier requires as input the current state $s_i$ and the suffix of the task plan $\langle \omega_i, \ldots, \omega_n \rangle$. Notably, both of these inputs vary in length across problems and even across steps within a problem— the number of steps in the task plan trivially affects its length, and

assuming that the continuous state set $S_t$ for each type $t$ is a real-valued vector space $\mathbb{R}^k$, the number of objects affects the dimensionality of the state. To deal with these variations, we use a transformer architecture (we could alternatively have used a recurrent neural net) that consumes tokens that represent state-action pairs, as illustrated in Figure 3. In principle, we could have two sets of tokens: one representing $s_i$ as a set of object states, and another representing the sequence of ground abstract actions in the task plan. However, it is unclear how a transformer would simultaneously handle the (ordered) sequence of actions and the (unordered) set of states.

Instead, we let each token be a pair $\langle s_j, \omega_j \rangle, 1 \le j \le n$. Such a token must contain information to identify the abstract lifted action $\bar{\omega}_j$, the objects that ground the action $o_1, \ldots, o_{m_{\bar{\omega}_j}}$, and the state $s_j$. Including the entirety of $s_j$ in the input would again require a varying-length representation. Since we will have multiple such tokens, we can restrict the representation of $s_j$ to the state of objects in the action, $s_{j,o_1}, \ldots, s_{j,o_{m_{\bar{\omega}_j}}}$. Once the current token is sequenced with the remaining tokens, the transformer will receive information about all objects that parameterize any

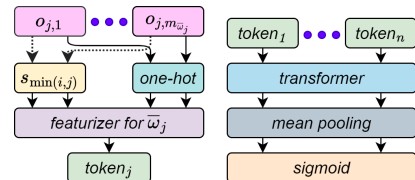

Figure 3: Architecture of the feasibility classifier $h$, supplied with the sequence of ground abstract actions $\omega_j = \bar{\omega}_j(o_{j,1}, \ldots, o_{j,m_{\bar{\omega}_j}}), 1 \le j \le n$ and refined continuous states $s_0, \ldots, s_i$.

action in the plan, which can be thought of as the set of objects relevant to the TAMP problem. To identify each object $o_l$, we use a one-hot encoding with a fixed maximum size $r_{\text{maxobj}}$—we randomize the one-hot encodings during training to avoid overfitting to the particular object instances in the training. One remaining challenge is that the number of objects $m_{\bar{\omega}}$ varies across actions. To produce a fixed-length token representation and introduce the identity of the action, we concatenate the object states and one-hot encodings and pass them through a lifted abstract action-specific *featurizer network* with a simple feedforward architecture [25] and ReLU activations [26], trained end-to-end with the transformer (details in Appendix C). For each ground abstract action that has already been refined $\omega_{1 \le j \le i}$, we use the state $s_j$ produced by the controller, while for subsequent actions we use the latest state $s_i$. Because featurizer inputs from such states may be drawn from different distributions than the ones produced from states generated directly by the controller (e.g., if object positions are encoded relative to the robot), we train separate featurizer networks for ground abstract actions $\omega_{j>i}$ that have not been refined.

### 4.3 Data Gathering

We assume access to a dataset $\mathcal{D}$ of demonstrations in the form of task plans, controller plans, and state sequences: $\langle \pi, p, \sigma \rangle$. Demonstrations can trivially be used as positive training data for the classifier: all parameters in the controller plan were sampled correctly, so any prefix of the controller plan $\langle \theta_1, \ldots, \theta_i \rangle$ and its associated sequence of states $\sigma = \langle s_0, \ldots, s_i \rangle$ should be accepted by our classifier. One (naïve) way to obtain neg-

```
1: function DATA-GATHERING(𝒟)
2:     h := Constant(1)
3:     𝒟₊, 𝒟₋ := {}, {}
4:     for l = max{n − 2 : ⟨π, σ⟩ ∈ 𝒟} … 0 do
5:         𝒟′ := {⟨π, σ⟩ ∈ 𝒟 : n − 1 ≥ l + 1}
6:         for random r_dp ⟨π, σ⟩ ∈ 𝒟′ do
7:             σ : ⟨s₀, …, sₙ⟩, π : ⟨ω₁, …, ωₙ⟩
8:             σ_pref := ⟨s₀, …, s_l⟩
9:             T := BT-h-CaptureSearchTree(σ_pref, π, 1.0, h)
10:            𝒟₊ ∪= {⟨π, σ_pref ⊕ ⟨ŝ⟩⟩ : ŝ ∈ SuccessfulTries(T)}
11:            𝒟₋ ∪= {⟨π, σ_pref ⊕ ⟨ŝ⟩⟩ : ŝ ∈ StuckTries(T)}
12:        h := TrainNeuralClassifier(𝒟₊, 𝒟₋, l)
13:    return h
```
Algorithm 3: Generating data for the Feasibility Classifier.

ative data is to run the backtracking search from Algorithm 1 on the same TAMP problems that we obtained demonstrations for. In the resulting search tree, from each node $s_0, \ldots, s_{n-1}$ on the successful path (except for the last one), there is a single branch that corresponds to a successful plan. All remaining branches failed to be refined, and because we want to find prefixes that are likely to be refined by backtracking with a fixed $r_{\text{iter}}$, these can be treated as unrefinable states. Concretely, we can take any prefix of the successful state sequence $\sigma = \langle s_0, \ldots, s_i \rangle$ and follow any application in the search tree of the subsequent controller $\mathfrak{c}_{\omega_{i+1}}$ with parameters $\hat{\theta} \neq \theta_{i+1}$ that differ from the successful controller plan. The resulting state $\hat{s} = \mathfrak{c}_{\omega_i}(\hat{\theta}, s_i)$ can be combined with $\sigma$ to produce

a negative data point for the given task plan $\pi$. This process is illustrated in Figure B.1. Note that it is imperative that the negative data point shares **all but the last state** $\hat{s}$ with a successful plan; otherwise, *any* state outside of the successful plan could have caused the failure. This implies that, to generate negative data points, backtracking must also find a successful motion plan. As explained in Section 3.2, this process takes exponentially long.

To alleviate this cost, we will use the fact that the backtracking search can also be started from any state $s_l$ on the successful sequence of states $\sigma = \langle s_0, \ldots, s_n \rangle$. We then alternate running classifier-aided backtracking (Algorithm 2) to generate negative data starting from $s_l$ and training a classifier using the obtained data to use for the next round of backtracking for data collection for a decreased value of $l$. The process is summarized in Algorithm 3. Initially, the feasibility classifier always outputs a confidence of 1 (Line 2). We then iterate over descending state prefix lengths, starting from $n - 2$ (Lines 4–12). For each task plan prefix length $l$, we select $r_{\text{dp}}$ viable data points (i.e., task plans of length $\geq l + 2$; Lines 5–6). For each data point, we run backtracking and return the entire search tree traversed by the algorithm (Line 9) and generate the feasibility classifier data (Lines 10–11). Note that we also generate positive data points in this step, because the samplers used for backtracking may generate a distribution shift from the initial demonstrations $\mathcal{D}$. We then train the new classifier (Line 12) and decrease the prefix length. In practice, we stop the search from a given prefix upon finding the first positive or negative data point.

## 5 Experiments

Our experimental evaluation aims to address *(Q1)* whether using backtracking with a feasibility classifier can meaningfully improve the speed of plan refinement, *(Q2)* whether the method generalizes to unseen plan lengths, *(Q3)* how well the method handles reduced initial demonstration dataset $\mathcal{D}$, and *(Q4)* whether the data collection, the backjumping, and the architecture of the classifier are necessary for the method to work. We run all our experiments with 8 different datasets of task plans and state sequences and test them on 50 test problems for each dataset. The evaluation metric is whether a method was able to find a solution motion plan within a timeout of 120s. See Appendix D complete success rate and timing data across all the domains. See Appendix A for additional details on the runtime, domains, baselines, and ablations.

### 5.1 Domains

We use 2D (Shelves, Donut, Statue) and 3D (Packing, Trays—simulated in PyBullet) quasi-static domains with collision checking only after each action (Figures 1 and 4). We provide abstract actions $\omega$ and controllers $\mathfrak{c}_\omega$ for each domain, and the learners must determine the continuous parameters $\theta_{\mathfrak{c}_\omega}$ that enable solving each problem. To simulate expert demonstrations to construct the dataset $\mathcal{D}$, we run SeSamE backtracking, using Fast Downward [7] as the task planner and ground-truth samplers to parameterize the controllers. For the 3D domains, we perform planning without simulated motion,

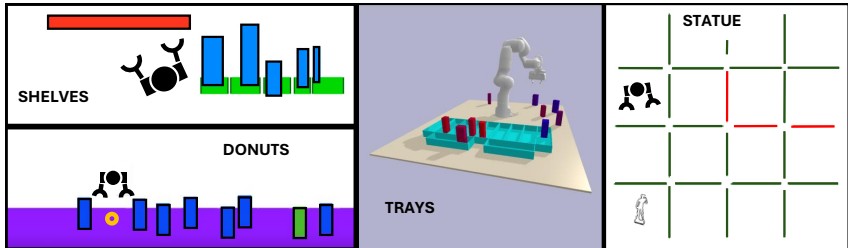

Figure 4: Evaluation domains. In *Shelves*, the blue boxes are placed on the green shelves and the red cover is placed on top or bottom. In *Donut*, the robot grasps the donut either from top or side, uses the topping machines (blue), and places it in the box or shelf (green). In *Trays*, the blocks should be placed in the different trays, which is not encoded at the task-plan level. In *Statue*, the robot should grasp the statue (horizontally or vertically) and move it to the top right corner; red doors cannot fit the statue grasped vertically, while narrow green doors cannot fit the statue grasped horizontally.

and then run bi-directional RRT [27] to validate the generated state sequences. Appendix F describes the steps necessary to run our method on a real robot.

- *Shelves:* Contains horizontally stacked shelves (training: 2–5, eval: 5–10) to be filled with boxes and then covered on the bottom or top (each box placement can cause cover placement failure).
- *Donut:* See Section 3.3. We use 1–3 topping machines for training and 3–6 for eval.
- *Statue:* Robot pick-and-place of a statue in a grid of rooms (side length—train: 2–4, eval: 4–8). Depending on the path from start to end, the statue has to be initially correctly picked up.
- *Packing:* A pick-and-place of blocks (training: 4–10, eval: 10) by a robot arm from a table to a box. Blocks are arranged in two rows; the back row has to be filled first due to collision.
- *Trays:* A pick-and-place of blocks (training: 4–8, eval: 8) by a robot arm from a table into a set of trays. Each tray should not have more than one block, but this can only be tested at the end.

## 5.2 Methods Evaluated

We evaluate the following methods.

- *Backtracking with Feasibility Classifier (Ours):* Our classifier-guided backtracking method from Section 4, using the diffusion-based samplers from [5]. There is one sampler for each lifted abstract action, and its input is the concatenation of the states of objects that ground the lifted action. Each sampler is a feedforward diffusion model trained on the data of the corresponding lifted action from the demonstrations in $\mathcal{D}$.
- *Myopic Diffusion Samplers (B1a):* The diffusion-based samplers used by our method, introduced in [5], used for general backtracking without a feasibility classifier.
- *Myopic Gaussian Samplers (B1b):* An alternative to B1a with similar parameterization, using the learned Gaussian-based rejection samplers from [6].
- *GNN Policy (B2):* A goal-conditioned graph neural net (GNN) [28] trained via behavior cloning.
- *Ablations:* We ablate the backjumping (A1) and the data-gathering procedure (A2, A3). See Appendix A for details.

## 5.3 Experimental Setups

To address *Q1–3*, we evaluated the baselines on the 2D domains. For *Q1* and *Q2*, we used a dataset $\mathcal{D}$ of 2000 data points, while for *Q3* we varied the dataset size between 500 and 2000 data points. For *Q3*, we used the minimum size for each test domain. For *Q2*, on 3D domains we varied the maximum number of blocks in the training data across the range described in Section 5.1. For *Q4*, we evaluated the ablations on the Pybullet domains, with a dataset of 2000 data points. For the remaining experimental data see Appendix D.

## 5.4 Results and Analysis

The results of Figure 5 demonstrate that our method is on average superior to all our baselines and ablations for all experimental setups. Additionally, it shows 40%+ improvement

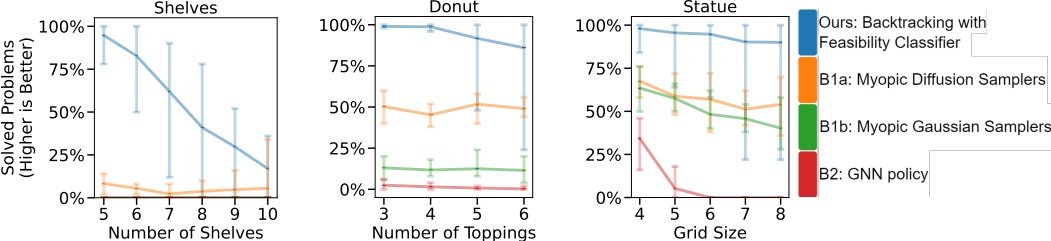

Figure 5: Generalization results on 2D domains across varying domain sizes. Our approach is the only one that solves nearly all problems in the minimum environment size, and generalizes to increasing sizes. Avg. across 8 seeds, ranges are min. and max, 0% plots are dropped for clarity.

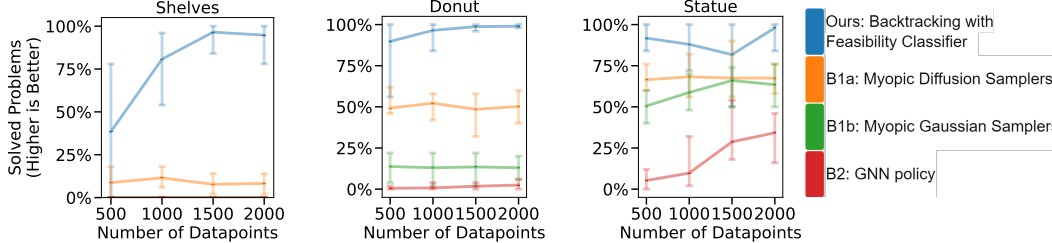

Figure 6: Data efficiency results on 2D domains across initial dataset sizes. Our approach solves most problems given the maximum initial dataset size, and outperforms baselines in smaller data regimes. Avg. across 8 seeds, ranges are min. and max, 0% plots are dropped for clarity.

in number of problems solved over the baselines on the non-generalization and non-reduced-data experiments. Thus, *Q1* can be answered affirmatively. For *Q2*, our method shows limited generalization for some domains. Its accuracy on the Shelves domain reduces with more shelves. One explanation is that the size of the cover grows with the number shelves, making this our only environment where an individual object's state is out-of-distribution for the networks. In addition, transformers show limited generalization to out-of-distribution input lengths [29], which may be a contributing factor.

The results in Figure 6 address *Q3*, showing superior performance in reduced data regimes. More initial training data improves performance.

Figure 7 contains results to answer *Q4*. Compared to our results, the data collection ablation *A2* and *A3* shows on average 25%-50% reduced performance on the Trays domain, which demonstrates that our bootstrapping approach for data collection is critical to the method's overall success. Simmilarly, ablating the backjumping (*A1*) reduces the method's performance by on average 50% reduced solution success rate on the Packing domain, which highlights the significance of that procedure in our method.

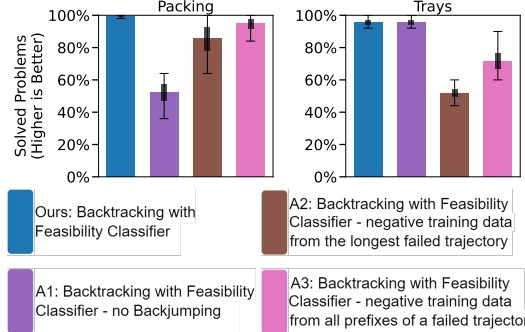

Figure 7: Ablations on the Packing and Trays domains. Assuming last action to be the wrong one (A2) reduces success rate. Backjumping (A1) and assuming all actions to be wrong in case of planning failure (A3) maintains success rate. Avg. across 8 seeds, error bars are stddev., ranges are min. and max.

## 6 Conclusion and Limitations

We proposed a solution to learn to account for long-horizon action dependencies in sampling-based bilevel planning. We showed experimentally that our approach substantially outperforms existing methods in problems requiring long-horizon reasoning. One limitation of our approach, common to most learning methods based on neural nets, is that using a small feasibility classifier network, or a small initial dataset of demonstrations for training, leads to degraded performance. The common solutions to use bigger networks or collect larger datasets can address this limitation. One challenge specific to our method is that data is labeled as positive or negative by our own data collection mechanism. If our sampling distributions $\psi$ are low-quality (which can happen in the case of learned distributions similar reasons of scale) or we attempt too few samples $r_{\text{iter}}$ at each step of backtracking, this can lead to noisy labels that complicate the training of the classifier. Increasing $r_{\text{iter}}$ can reduce this labeling noise. Our choice to use transformer architectures carries the drawback that generalization to longer inputs (in this case, longer plans) is limited (see [29]). Applying regularization techniques during training can mitigate this challenge. Other directions to improve our approach include limiting the distance of action dependencies or collecting data only every $k$-th prefix length to accelerate data collection, or using classifier-based guidance [30] to more efficiently sample from diffusion models.

**Acknowledgments**

We thank Aditya Agarwal for initial discussions on the feasibility classifier architecture. The research of J. Mendez-Mendez is funded by an MIT-IBM Distinguished Postdoctoral Fellowship. We gratefully acknowledge support from NSF grant 2214177; from AFOSR grant FA9550-22-1-0249; from ONR MURI grant N00014-22-1-2740; from ARO grant W911NF-23-1-0034; from the MIT-IBM Watson AI Lab; from the MIT Quest for Intelligence; and from the Boston Dynamics Artificial Intelligence Institute. The experimental data was gathered using the MIT Supercloud [31].

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

# Appendices to
# "Learning Long-Horizon Action Dependencies in Sampling-Based Bilevel Planning"

**Bartłomiej Cieślar**[1], **Leslie Kaelbling**[1], **Tomás Lozano-Pérez**[1], and **Jorge Mendez-Mendez**[2]

[1]CSAIL MIT            [2]Stony Brook University

{bcieslar,lpk,tlp}@csail.mit.edu   jorge.mendezmendez@stonybrook.edu

## A    Additional Experimental Details

For the purpose of reproducibility, this appendix contains additional details of the runtime, domains, baselines, and ablations for the experiments in Section 5 in the main paper.

### A.1    Runtime

All experiments were run on Ubuntu 22.04.3 using 20 cores of Intel Xeon Gold 6248 and a 32GB Nvidia Volta V100. We use 30 threads to collect the data for our method and its ablations in parallel. For the 3D domains, we use PyBullet for collision checking and rendering.

### A.2    Detailed Domain Descriptions

This section provides additional details of the domains.

- *Shelves Domain:* Contains horizontally stacked shelves to be filled with boxes and then covered on the bottom or top (as specified by the goal). The boxes are always taller than the shelves, requiring long-horizon reasoning of how to avoid them interfering with the cover. Each box placement introduces a possible failure. The cover must be placed after the boxes are on the shelves. The lifted abstract actions are "Move Box", "Move Cover to Top" and "Move Cover to Bottom". All lifted abstract actions share a single controller, whose parameters are some $\langle x, y \rangle$ absolute position of a point on the moved and on the target object, and the $\langle x, y \rangle$ offset with respect to the target destination to place the moved object on. The positions of the objects are in absolute coordinates, and the objects cannot be rotated.

- *Donut Domain:* A robot picks a donut, adds toppings using topping machines, and places the donut in a box or on a shelf (depending on the goal). The donut must be grasped from the top to place it in the box, and from the side for the shelf; this is the main point of failure. The lifted abstract actions are "Move Robot", "Grasp Donut" "Place Donut in Box", "Place Donut on Shelf" and "Add Topping to Donut". The movement, grasp position and placement position in the abstract action controllers are parameterized by the $\langle x, y \rangle$ displacement. All other controller parameters are determined using binary decisions based on a threshold on a real value (e.g., either a top or side grasp, or which topping to add). The topping machines cannot be used if they're too far away from the robot; a similar restriction applies to the grasping and placement actions. The positions of the objects are relative to the robot, and the objects cannot be rotated. There are 10 varieties of possible toppings in the goals.

- *Statue Domain:* A robot traverses a grid of rooms connected by doors to pick up a statue from the bottom-left corner and place it in the top-right corner. The statue can be carried horizontally or vertically. All possible paths to goal go through one door that can only fit the statue carried one way. The main point of failure is the statue grasp, but failures are only apparent upon reasoning about the entire task plan, and not just the goal. The lifted abstract actions are "Go through Door", "Go through Door with Statue", "Grab Statue" and "Place Statue". The movement controller parameterization specifies the offset $\langle x, y \rangle$ that the robot moves by, and the grab controller specifies a thresholded value for the orientation of the statue (horizontal, vertical) in the 3D axis from the front of the robot (not perpendicular to the world). The positions of objects are in absolute coordinates, and (other than the horizontal/vertical rotations of the statue) the objects cannot be rotated.

In training problems, the positions of objects are randomly offset to keep them in distribution for the larger grids of rooms in test problems.

- *Packing Domain:* A Franka Panda arm grabs blocks from the table and places them in a box, open from the front and facing the robot. The blocks only fit if placed in two rows, so the first blocks must be placed in the back to make room for the subsequent ones in the front. The blocks initially lie down scattered across the table. The only lifted abstract action is "Place Block". The controller is parameterized by where along the block to grasp it and where to place the block (upright) relative to the center of the box. The positions of objects are in absolute coordinates, and the rotations are in quaternions.

- *Trays Domain:* A Franka Panda arm grabs blocks from the table and places them in one of 9 trays. The goal is to place only one block in each tray, but the relationship between blocks and which trays they are placed into is not encoded at the task-plan level. The blocks are initially upright scattered across the table. The lifted abstract actions are "Move Block" and a *dummy* "Check Trays" action that confirms (at the end of the plan) if the blocks are in the target configuration— this is necessary for the task planner to find a goal-reaching plan, but prevents the movement actions from knowing the precise target placements. The controller is parameterized by where to place the block (upright) relative to the center of the tray and a one-hot encoding of the tray. The positions of objects are in absolute coordinates, and the rotations are in quaternions.

### A.3 Backjumping and Data Collection Ablations

This section describes the ablations used to validate the need for each element of our approach.

- *No backjumping (A1):* To study the impact of backjumping on our method, this ablation disables backjumping, instead always backtracking only one step. The ablation uses exactly the same trained networks as our full approach.

- *Negative training data from the longest failed trajectory (A2):* To assess the importance of our data collection scheme, we train our method by generating positive training data from the prefixes of controller plans from the demonstrations dataset $\mathcal{D}$ and negative training data from the longest failed refinements of attempts at running backtracking on the problems from $\mathcal{D}$.

- *Negative training data from all prefixes of a failed trajectory (A3):* This ablation is similar to *A2*, but instead generates negative training data from all prefixes of the longest failed refinement.

### A.4 Training the Baselines

Table 1 presents the hyperparameters used when evaluating the baselines and the diffusion-based samplers used in our method (classifier hyperparameters are included in Appendix C).

The hyperparameters of each method were tuned based on its prediction accuracy on the Shelves domain, using a held-out validation dataset of 20% of the data points. We picked the hyperparameters with the lowest validation loss for each method. The SeSamE-based methods (including our own) set $r_{\text{iter}} = 20$ for all domains. Our data collection method from Section 4.3 collects $r_{\text{dp}} = 4000$ data points per the iteration of the data collection loop. To ensure a fair comparison, the Gaussian and diffusion samplers were trained for a comparable amount of time, and the GNN baseline was trained for a similar amount of time as our method (including the data collection procedure).

## B  Data Gathering Illustrative Example

Figure B.1 illustrates an example search tree that the backtracking algorithm could produce during the data collection described in Section 4.3.

## C  Network Training Setup

In this appendix, for the purposes of reimplementation and reproducibility, we describe the training setup for our transformer-based classifier. We use an encoder-only transformer.We select the hyperparameters for the classifier, summarized in Table 2, based on the accuracy on the final iteration of training in the data collection algorithm from Section 4.3 on the Shelves environment—runs that

Table 1: Hyperparameters for the baselines and samplers for our method

| Method | Hyperparameter | Value |
|---|---|---|
| Diffusion sampler | number of training iterations | 10000 |
| | number of diffusion timesteps | 100 |
| | hidden layer sizes | $2 \times 512$ |
| | learning rate | $1e-4$ |
| Gaussian sampler | regressor hidden sizes | $1024 \times 2$ |
| | classifier hidden sizes | $128 \times 2$ |
| | number of training terations | 20000 |
| | learning rate | $1e-3$ |
| GNN | number of epochs | 1600 |
| | number of message passings | 3 |
| | hidden sizes (encoders, models, and decoders) | $1 \times 512$ |
| | learning rate | $1e-4$ |

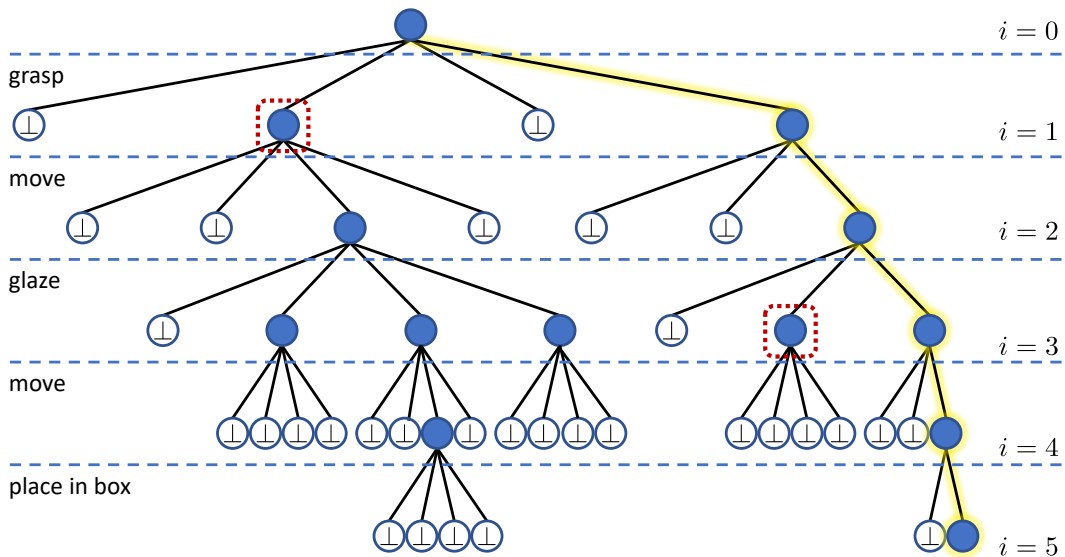

Figure B.1: Example data collection search graph. This is the subgraph of the refinability graph described in Section 3.3 found via backtracking search on the Donut domain, using $r_{\mathrm{dp}} = 4$. Rows separated by dashed lines represent layers of the graph (as labeled by $i$), filled nodes represent continuous states $s_i$ found during backtracking search, and empty nodes labeled by $\perp$ indicate that the sampled parameters resulted in controller failure. The path highlighted in yellow corresponds to a successful execution, so all nodes and corresponding prefixes of the plan are positive samples for $h$. The two nodes boxed by red dotted lines exemplify the negative samples that lead to refinement failure, as described in Section 4.3.

caused the data collection to take prohibitively long (e.g., because accuracy in early iterations was too low) were terminated early and discarded.

We optimize a standard binary cross-entropy loss with the Adam optimizer, over randomly drawn mini-batches. In addition to the output of the featurizer network, for each token we concatenate a one-hot encoding of which featurizer network produced it (identifying the lifted abstract action and whether the state was produced by its controller, as explained in Section 4.1), and a binary flag to indicate whether the corresponding ground abstract action was the latest one to be refined (i.e., the action $\omega_j$ such that $j = i$). These additional inputs aid the network with locating the key information in the sequence of tokens. As is standard practice in the literature, we also concatenate the sinusoidal positional encodingto each token. As described in [29], we offset the positions of the tokens by a random value to improve out-of-distribution generalization with respect to the length of the task plan. Before passing the token to the transformer, we pass it through a linear map to ensure

Table 2: Feasibility classifier training hyperparameters

| Hyperparameter | Value |
|---|---|
| Model learning rate (without the transformer) | $1e-4$ |
| Transformer learning rate | $1e-5$ |
| Number of training iterations | 5000 |
| Batch size | 4000 |
| Featurizer network hidden layer sizes | $2 \times 256$ |
| Feautrizer network output size | 256 |
| Sinusoidal embedding dimensionality | 128 |
| Sinusoidal embedding base | 130 |
| Transformer token width | 128 |
| Transformer feedforward block hidden size | 512 |
| Number of transformer heads | 8 |
| Number of transformer residual blocks | 4 |

matching dimensionality. Our transformer then uses multi-head attention: the token is split into a number of transformer heads, each processing a chunk of the mapped token.

## D  Additional Experimental Results

In this appendix, we present the experimental results not included in Section 5.4. The timings only consider the tasks for which each baseline succeeded, which is why on the 2D domains our method sometimes exhibits a higher runtime than the baselines. The timings for the GNN baseline (B2) were not included because the policy is designed to compute a single solution; if the solution works, it succeeds, and otherwise it immediately fails. In consequence, when it does succeed, it is much faster ($\sim 1000\times$) than our approach.

## E  Simplified Explanation of Our Contribution

SeSamE approaches [5, 6] methods for task and motion planning work by first creating a task plan of high-level operators (such as *grab cup* or *place the cup on the table*) and finding controller parameters for the motion controllers associated with those operators (the motion plan). The task plans are commonly found using common search algorithms such as Fast Downward [7] with A* [24]. The controller parameters are found by their recursive sampling step-by-step for each of the motion controllers associated with the operators in the task plan using the *feasibility classifier*. Our contribution centers around preemptively rejecting the controller parameters that are likely to result in downstream search failures (e.g., when an incorrectly grasped Donut cannot be placed into a box due to collision of the arm with the box). See Section 4.1 for the details of the recursive controller parameter discovery process.

We condition the feasibility classifier on states generated after running each controller in sequence. The states are then passed through *featurizer networks*, which are a simple feedforward architecture [25] with a ReLU activation [26], to form a sequence of tokens. These tokens are then concatenated with sinusoidal positional embeddings and passed through a transformer architecture [32]. The tokens produced by a transformer are then averaged together and passed through a linear layer with a sigmoid activation [33] to produce the final confidence of whether further controller parameter discovery is possible. The *featurizer networks*, transformer and final linear layer are trained jointly in a supervised manner. See Section 4.1 for the rationale behind the specific choice of the feasibility classifier architecture.

To gather the data for training the feasibility classifier we use the fact that, during the controller parameter search, each controller parameter for which the previous steps of recursive search succeed but for which subsequent search fails can be used as negative training datapoints. Likewise, each controller parameter for which the subsequent search succeeds can be used as a positive training datapoint. We also notice that, given a successful motion plan, the controller parameter search can be restarted at any point in that plan to produce the aforementioned datapoints. The challenge to

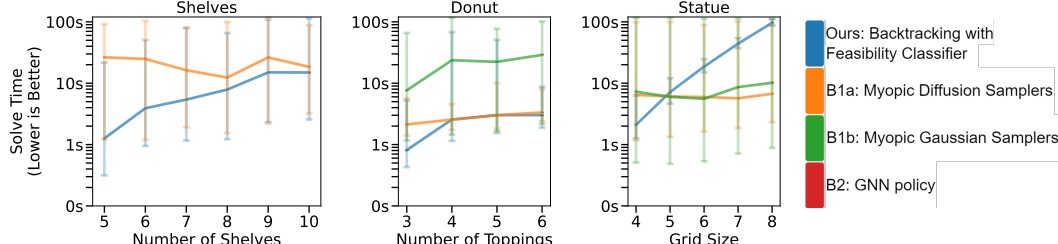

Figure D.2: Solve times for the generalization experiments on 2D domains across varying domain sizes. Other than the Statue domain, our approach is faster than the sampler-based baselines (B1a and B1b) across all environment sizes. We omit the GNN baseline (B2) because it does not do planning and therefore is trivially faster than planning approaches. Averaged across 8 seeds, ranges represent mininum and maximum values. Note that 0% plots are dropped for clarity.

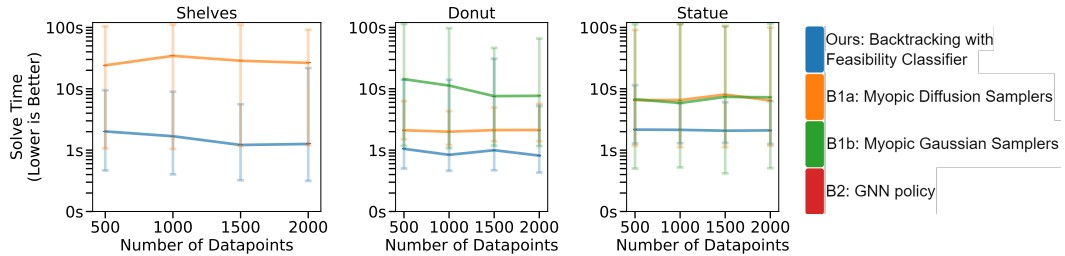

Figure D.3: Solve times for the data efficiency experiments on 2D domains across initial dataset sizes. Our approach is on average faster than the sampler-based baselines (B1a and B1b) across all dataset sizes. We omit the GNN baseline (B2) because it does not do planning and therefore is trivially faster than planning approaches. Averaged across 8 seeds, ranges represent minimum and maximum values. Note that 0% plots are dropped for clarity.

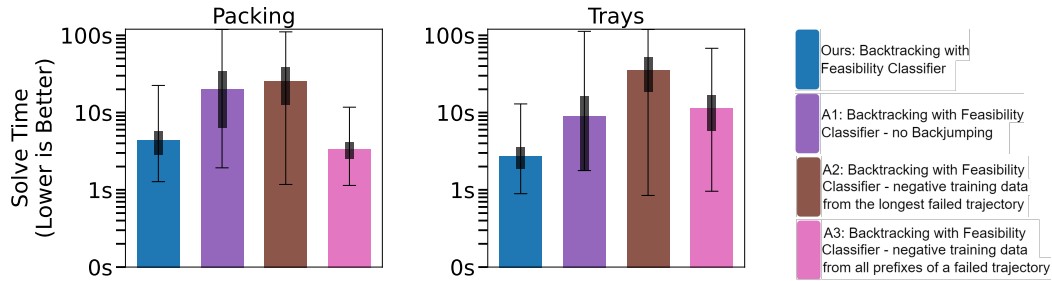

Figure D.4: Solve times for generalization (left) and data efficiency (right) results on 3D domains across varying domain sizes. Our approach is consistently over $3\times$ faster than ablations A1 and A2; the ablation A3 is comparably fast on the Packing domain, but over $3\times$ slower on the Trays domain. Avg. across 8 seeds, ranges are min. and max, 0% plots are dropped for clarity.

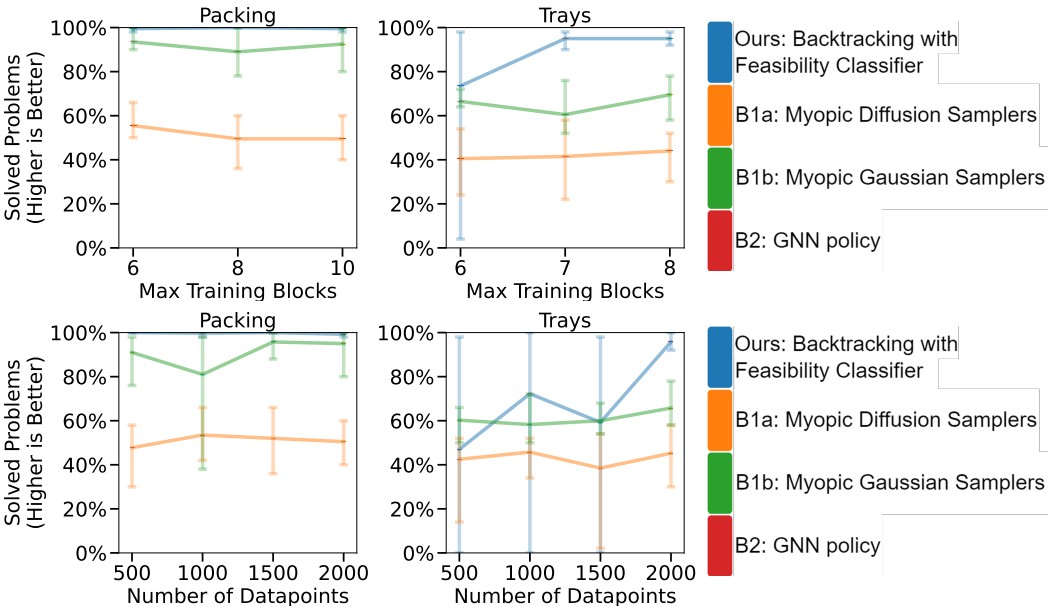

Figure D.5: Results on generalization (top) and data efficiency (bottom) of our method and baselines on 3D domains across varying maximum number of blocks during training and initial dataset sizes. Our approach shows a higher success rate on almost all experimental setups. Avg. across 8 seeds, ranges are min. and max, 0% plots are dropped for clarity.

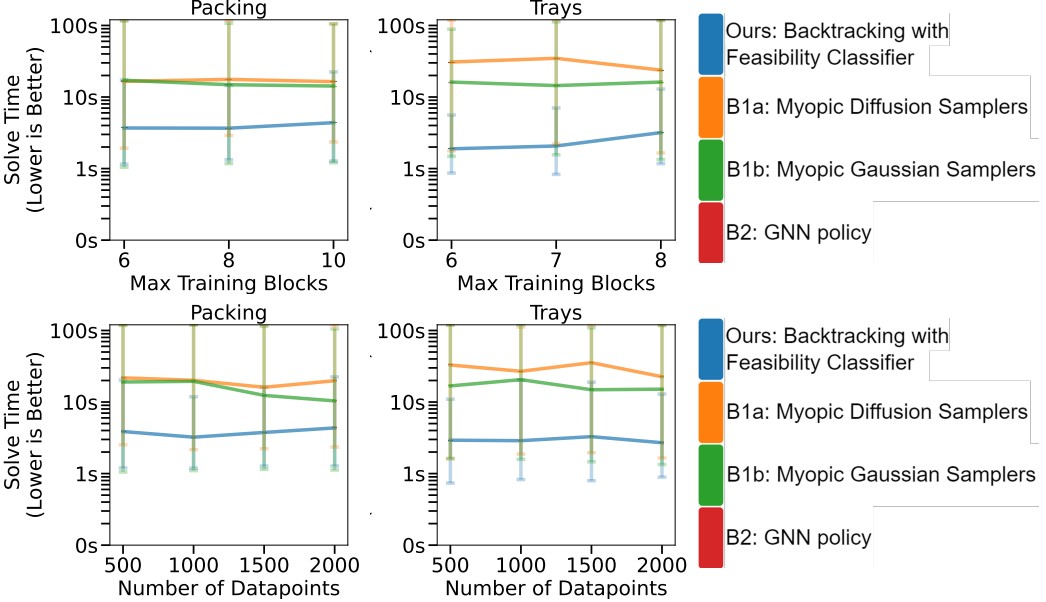

Figure D.6: Solve times for generalization (top) and data efficiency (bottom) results on 3D domains across varying domain sizes. Our approach is consistently over $5\times$ faster than baselines B1a and B1b. Avg. across 8 seeds, ranges are min. and max, 0% plots are dropped for clarity.

overcome is that, without the use of an already functioning feasibility classifier, this datapoint collection procedure is prohibitively long. Thus, in our data collection algorithm, given a set of already found motion plans, we step-by-step restart the controller parameter search from progressively earlier points in the motion plans, train our feasibility classifier after each such step, and use it to speed up the search for subsequent data collection steps. See Appendix B for an illustrated example of this process and Section 4.3 for a more detailed description of the algorithm.

## F   Discussion of running the method on a real world setup

While our experiments in Section 5 in the main paper focused on simulated domains, our approach could feasibly be used on a real robot under certain constraints. In summary, other than a faithful simulator of the real world scene, we require a source of data to train the feasibility classifier, which **need not be realistic**, just representative of what plans will be run **on the simulator** during evaluation. Specifically, the following components would be crucial to add to our system before running it on real hardware:

- *A simulator* synchronized with the real robot. The challenge here is making sure the positions of objects are well represented in the simulator (since the robot joint positions can usually be accurately measures). Assuming the objects used in the domain are known, or can be extracted from somewhere, locating them can be achieved using e.g., Aruco tags [34], using ICP [35].

- *Data of successful trajectories*, which can be collected by running the backtracking search without the heuristic in the simulator (baseline B1a from Section 5.2) in a parallelized manner. While this method does require some source of controller parameter samplers, which in our case are learned through Imitation Learning, they require much less trajectory data to train, as shown in [5]. Alternatively, the data can be collected by having a human click through a trajectory for the robot to follow.

- *Real-world abstract action controllers* can be achieved by utilizing the synchronization of the simulator with the real world and using the controllers used for planning in the simulated environment. Importantly, the plan found during motion planning with the simulator would have to be executed in an open-loop manner, i.e. under the assumption of no catastrophic execution failure (such as the robot dropping an object). Should we want to handle a case of such a failure, a replanning procedure would have to be introduced into the task planner and we would have to collect data for such failure recovery during the successful plan search.

