# OpenReview forum: "Learning Long-Horizon Action Dependencies in Sampling-Based Bilevel Planning"
_robot-learning.org/CoRL/2024/Conference — CoRL 2024_

### Official Review · Reviewer_5rxt · 2024-07-21
**Important and novel approach on long-horizon dependencies in TAMP.  Needs improvement in related work and experimental evaluation.**

**Originality:** 3
**Technical Quality:** 4
**Clarity Of Presentation:** 4
**Potential Impact:** 3
**Recommendation:** 3
**Confidence:** 4

**Review:**

Strength:

The paper considers the important and relevant problem of long-horizon dependencies TAMP planning. The authors compared against 3 different methods. The authors provided a detailed and clear explanation on the data generation method  and how to include the classifier in the TAMP approach.

Weakness:

More detailed clarity in TAMP formulation section is needed.  The authors need to include relevant work in related section. Experimental evaluation need to be focused on 3D problems.  Details are provided below:

1) The author should consider other traditional and learning based method on culprit detection problem and guided search in TAMP such as:-
Lagriffoul F, Andres B. Combining task and motion planning: A culprit detection problem. The International Journal of Robotics Research. 2016;35(8):890-927. doi:10.1177/0278364915619022
Wells, Andrew M., et al. "Learning feasibility for task and motion planning in tabletop environments." IEEE robotics and automation letters 4.2 (2019): 1255-1262.
C. V. Braun, J. Ortiz-Haro, M. Toussaint and O. S. Oguz, "RHH-LGP: Receding Horizon And Heuristics-Based Logic-Geometric Programming For Task And Motion Planning," 2022 IEEE/RSJ International Conference on Intelligent Robots and Systems (IROS), Kyoto, Japan, 2022, pp. 13761-13768, doi: 10.1109/IROS47612.2022.9981797. keywords: {Legged locomotion;Heuristic algorithms;Decision making;Kinematics;Programming;Robustness;Planning},
M. Khodeir, A. Sonwane, R. Hari and F. Shkurti, "Policy-Guided Lazy Search with Feedback for Task and Motion Planning," 2023 IEEE International Conference on Robotics and Automation (ICRA), London, United Kingdom, 2023, pp. 3743-3749, doi: 10.1109/ICRA48891.2023.10161109. keywords: {Adaptation models;Runtime;Codes;Automation;Search problems;Manipulators;Skeleton},

2) Section 3.1 need more explanations and definitions with examples. For example, it is difficult to understand the difference of lifted abstract actions and ground abstract actions.

3) Although the authors stated RNN can be used as the feasibility classifier, an explanation behind using transformer as the underlying architecture will be useful.

4) The experiments and comparison are conducted primarily with 2D environments. Most of existing TAMP problems are in 3D. The authors should provide the experimental evaluation focused on the results from 3D environments. What are the DOF of the robots in 2D environments?

5) The authors should consider benchmark TAMP problems from: https://pypi.org/project/guided-tamp-benchmark/
and http://tampbenchmark.aass.oru.se/index.php?title=Main_Page

**Quality Of The Limitations Section:**

3

**Questions For Rebuttal:**

1) The author should consider other traditional and learning based method on culprit detection problem and guided search in TAMP such as:-
Lagriffoul F, Andres B. Combining task and motion planning: A culprit detection problem. The International Journal of Robotics Research. 2016;35(8):890-927. doi:10.1177/0278364915619022
Wells, Andrew M., et al. "Learning feasibility for task and motion planning in tabletop environments." IEEE robotics and automation letters 4.2 (2019): 1255-1262.
C. V. Braun, J. Ortiz-Haro, M. Toussaint and O. S. Oguz, "RHH-LGP: Receding Horizon And Heuristics-Based Logic-Geometric Programming For Task And Motion Planning," 2022 IEEE/RSJ International Conference on Intelligent Robots and Systems (IROS), Kyoto, Japan, 2022, pp. 13761-13768, doi: 10.1109/IROS47612.2022.9981797. keywords: {Legged locomotion;Heuristic algorithms;Decision making;Kinematics;Programming;Robustness;Planning},
M. Khodeir, A. Sonwane, R. Hari and F. Shkurti, "Policy-Guided Lazy Search with Feedback for Task and Motion Planning," 2023 IEEE International Conference on Robotics and Automation (ICRA), London, United Kingdom, 2023, pp. 3743-3749, doi: 10.1109/ICRA48891.2023.10161109. keywords: {Adaptation models;Runtime;Codes;Automation;Search problems;Manipulators;Skeleton},

2) Although the authors stated RNN can be used as the feasibility classifier, an explanation  behind using transformer as the underlying architecture will be useful.

3) The experiments and comparison are conducted primarily with 2D environments. Most of existing TAMP problems are in 3D. The authors should provide the experimental evaluation focused  on the results from 3D environments.

4) What are the DOF of the robot in 2D environments?

5) The authors should consider benchmark TAMP problems from: https://pypi.org/project/guided-tamp-benchmark/
and http://tampbenchmark.aass.oru.se/index.php?title=Main_Page

**Robotics Focus:**

3

**Summary Of Paper:**

The paper presents an approach of handling long-horizon action dependencies in TAMP problems.  The proposed method uses a transformer based feasbility classifier that determines whether a given task plan can be refined from a current input state. The classifier is also used  in alternatively to generate negative data set in between its' training. The confidence  returned by the classifier is used to backtrack to the previous step on failed searches. Experiments are conducted on 2D and 3D environment and compared against diffusion based  and GNN based models in terms of success rate. The proposed method shows superior  performance in the experimental environments.

**Summary Of Recommendation:**

The paper shows a novel approach in TAMP with promising results in 2D. Needs improvement in experimental evaluation and related work.

---

### Official Review · Reviewer_dUuw · 2024-07-21
**Learning search heuristics for sampling-based TAMP**

**Originality:** 4
**Technical Quality:** 5
**Clarity Of Presentation:** 4
**Potential Impact:** 3
**Recommendation:** 3
**Confidence:** 5

**Review:**

The paper makes a solid contribution demonstrating the applicability of supervised learning methods to learning heuristics for sampling-based TAMP problems. The paper can be improved in the following ways.

1, The paper presentation can be improved to make it more friendly to people who are less familiar with the literature of TAMP (sampling-based or not) and backtracking search (with heuristics or not). The "packing" example shown in Figure 1 can be revised to serve such purposes.

2, It's not surprising to see that one can learn heuristics for guiding the backtracking search (for TAMP or other tasks) -- generally it's expected one can leverage supervised learning methods to improve search efficiency. The unknown is on the generalization of the learned heuristics (Q2 in Section 5 on Experiments). Unfortunately, the experiments only evaluated the generalization in domains of different sizes. Could the proposed approach generalize to novel objects, novel requests, novel tasks, different actions, and even different domains?

3, This is a minor comment. Figure 2 is helpful as it highlights the point of the learned heuristic function but still could be more intuitive. Some visuals on a real robot (or simulation) would be very nice. The current version requires some imagination to clearly get the point.

4, It seems the paper doesn't include a real-robot component.

**Quality Of The Limitations Section:**

3

**Questions For Rebuttal:**

Given the dataset, couldn't one directly learn a TAMP planner? What's preventing us from doing so? Is it still necessary to use a fast-forward planner?

Is it the case that the generalization evaluation was only performed in the same domain (where the training data was collected and where the experiments were performed)? What's its performance in generalization beyond that?

Is there a reason that the proposed approach is difficult to be realized on a real robot? Evaluations or demonstrations would be helpful.

**Robotics Focus:**

3

**Summary Of Paper:**

This paper aims to develop a learning approach for improving the efficiency of solving sampling-based task and motion planning (TAMP) problems. The motivation of this proposed approach is that backtracking search is commonly used in resolving dependencies in sampling-TAMP, but backtracking is too slow for long-horizon TAMP problems. This paper assumes access to a dataset of triples, each including a task plan, motion plan and state sequence. Learning from such a dataset using Transformers produces a heuristic function for guiding the backtracking search. Results show improvements over baselines that use diffusion samplers and Gaussian samplers, as well as baselines of GNN policies. Three household domains were used in experiments.

**Summary Of Recommendation:**

The paper presents a solid contribution to the sampling-based TAMP literature. The experiments can be improved by evaluating its generalization capability using different tasks and on real robots.

---

### Official Review · Reviewer_KAhV · 2024-07-31

**Originality:** 3
**Technical Quality:** 4
**Clarity Of Presentation:** 4
**Potential Impact:** 3
**Recommendation:** 3
**Confidence:** 4

**Review:**

Positives:
- The problem the paper addresses is important and challenging.
- The paper is well-written, except for a few parts (see writing comments).
- The biggest positive for me is that the computational problem is grounded in a popular existing approach (SeSamE-based methods) and is motivated in a rigorous manner.
- I appreciate the automated voice used in the video very much as it strengthens the double blind review process

Negatives:
- There are no hardware experiments in the paper and this is a significant concern for me. A lot of TAMP research heavily relies on accurate simulation of real-world perception and physics. Further, this paper also heavily relies on data collection, which is performed in simulation as well. Several issues come up with data collection and TAMP in the real world; sometimes they are pervasive enough that you might have to throw away some of your rigor. It would have been preferred to begin with a focus on applying the approach to real-world, 3D environments as far as possible, even if at the expense of making technical assumptions for these experiments or covering fewer domains. I would like to see that the authors have thoroughly thought about this in their response to my first question for rebuttal and have a realistic plan for hardware experiments.

Writing comments:
- There is a critical reference to the domain from Section 5 in an involved explanation in Section 3: I think it is perfectly reasonable to describe the domain in Section 3 (mentioning that we use an example to ground understanding of the approach) and refer to *it* in Section 5. This will make Section 3 much easier to read.
- The notation for the TAMP problem description can be made a lot more readable by utilizing diagrams to depict the notions of continuous and abstract states, grounding, etc. Consider using [this blog post](https://lis.csail.mit.edu/bilevel-planning-for-robots-an-illustrated-introduction/) as a reference.
- Typically with papers heavy on notation, I think it is useful to have a “TL;DR” or an “In a Nutshell” section that uses minimal notation and terminology but still explains the core algorithmic contribution of the approach. I would appreciate adding such a section before the Problem Formulation. I understand that there is limited space and a trade-off here and this is only a suggestion. As a solution, consider moving either the results on the 2D domains or some of the result analysis to Supplementary material.

Note: My score for review confidence is a 4 solely because I have not verified in minute detail the math in Section 3.3 and correctness of Algorithm 2.

**Quality Of The Limitations Section:**

3

**Questions For Rebuttal:**

1. Can you please share in detail your thoughts on moving towards on-robot deployment of your approach? The response to this question does not need to be *long*; just thorough. For that, it might help to make sure that your response:
    - is grounded within the specific context of your paper as far as possible (and does not talk about addressing problems for extremely *general* approaches)
   - includes proposing a few concrete next steps (be it “engineering” or “research”)
   - includes thoughts about the data collection
   - includes references to support any claims where applicable
2. The problem of informing the “high level” about uncertainty in the “low level” is a classic and interesting problem with hierarchical model-based approaches. Naturally, researchers outside of the TAMP community have also looked into versions of this problem but use different words such as “control level” and “planning level”. For instance, consider the paper [“Operating with Inaccurate Models by Integrating Control-Level Discrepancy Information into Planning” by Ratner et al.](https://ieeexplore.ieee.org/document/10161389) Please briefly talk about how this work relates to yours and what the trade-offs are. Please also include a shorter version of your response to this question in the Related Work section.
3. The work heavily relies on data collection. Can you please briefly comment on the implications on generalization of your approach to unseen domains?
4. Further, the paper mentioned in question (2.) above uses only data observed during execution. Please also briefly comment on the trade-offs due to this distinction in their approach and yours.
5. How does your work relate to / overlap with work on learning symbols, such as initiation set classifiers? For an example of a paper in this area, see ["Learning Symbolic Representations for Planning with Parameterized Skills" by Ames at al.](https://ieeexplore.ieee.org/abstract/document/8594313)

**Robotics Focus:**

3

**Summary Of Paper:**

Bilevel planning approaches for long-horizon tasks that use sampling and backtracking to account for continuous-level failures typically require handcrafted sampling distributions for valid continuous samples. This is intractable for seemingly simple problems. The authors propose using data of successful and failed plans to learn a model (transformer) that predicts such a valid set to improve the efficiency of backtracking in these methods. The paper contains promising results in simulation.

**Summary Of Recommendation:**

The paper addresses an important and challenging problem relevant to hierarchical model-based planning methods, motivates it rigorously, and grounds it in popular TAMP approaches. However, due to no hardware experiments, my recommendation is "weak accept". Depending on responses to rebuttal questions, I could lean towards a "strong accept" or a "weak reject".

---

### Author Rebuttal · Authors · 2024-08-14

We wish to thank all the reviewers for thorough and helpful evaluations. In the responses, we:
 - describe a pipeline that would allow our method to be run on a real setup
 - discuss the scope of generalization of our method
 - reason about using a transformer-based instead of an RNN-based architecture for the feasibility classifier
 - discuss the relationship of our method to the works highlighted by the reviewers.

We also provide an amended version of our manuscript based on the reviewers' suggestions. We highlight all changes to the original paper in **red text**. These include:
 - a version of Section 3.1 expanded with additional real-world examples of the concepts introduced
 - moving the description of the Donut environment to Section 3.3, since it is used in the mathematical explanation of the setting of foresight and the reasoning behind the feasibility classifier
 - related work highlighted by the reviewers
 - an appendix with a simplified explanation of our contribution
 - an appendix with an outline of the real-world pipeline

---

### Decision · Program_Chairs · 2024-09-04

**Decision:**

Accept

**Comment:**

The reviewers found many positive aspects of the paper, but also clearly articulated some questions for the authors.
Thank you for your responses.